# Serum Biochemistry Parameters of the Endangered Miranda’s Donkey Breed: Reference Intervals and the Influence of Gender and Age

**DOI:** 10.3390/ani14050805

**Published:** 2024-03-05

**Authors:** Grasiene Silva, Ana C. Silvestre-Ferreira, Belen Leiva, Felisbina L. Queiroga

**Affiliations:** 1Departamento de Ciências Veterinárias, Universidade de Trás-os-Montes e Alto Douro (UTAD), 5000-801 Vila Real, Portugal; 2Centro de Ciência Animal e Veterinária (CECAV), Universidade de Trás-os-Montes e Alto Douro (UTAD), 5000-801 Vila Real, Portugal; 3Laboratório Associado para a Ciência Animal e Veterinária–AL4AnimalS, 1300-477 Lisboa, Portugal; 4AEPGA-Associação para o Estudo e Proteção Gado Asinino, M. Largo da Igreja, n° 48, 5225-011 Atenor, Portugal; 5Centre for the Research and Technology of Agro-Environmental and Biological Sciences (CITAB), 5000-801 Vila Real, Portugal

**Keywords:** serum biochemistry, Miranda’s donkey, reference intervals

## Abstract

**Simple Summary:**

Miranda’s donkey is a breed from the north of Portugal, considered endangered. Knowledge of its physiological and pathological conditions is essential to assist in the conservation of this breed. The objective of our study was to determine reference intervals for twenty-one blood biochemistry parameters and to evaluate the influence of gender and age on these parameters. Several parameters were influenced by gender and age, emphasizing the importance of taking these factors into account during the interpretation of clinical results by veterinarians. The RIs described here can be used to evaluate and monitor the health status of animals and herds, thus contributing to the preservation of this breed.

**Abstract:**

Miranda’s donkey is an endangered, autochthone breed from Northern Portugal. Understanding the physiological and pathological conditions of Miranda’s donkey is crucial for the conservation of the breed. Our study aimed to establish reference intervals (RIs) for blood biochemistry parameters and to evaluate the influence of gender and age on these parameters. Blood samples from 75 clinically healthy animals were analyzed for 21 blood biochemistry parameters using Respons^®^ 920 and Start^®^ 4-Diagnostica-Stago. RIs were calculated according to the ASVCP guidelines, utilizing Reference Advisor software V. 2.1 and the statistical program SPSS version 29 to analyze the effects of gender and age. Significant gender-related differences (*p* < 0.05) were observed in cholesterol, chlorine, creatinine (CREA), glucose (GLU), sodium, and triglycerides (with higher values in females) and in aspartate aminotransferase, creatine phosphokinase (CK), gamma-glutamyl transferase, potassium, magnesium (Mg), and urea (with higher values in males). Age-related differences (*p* < 0.05) were noted for alkaline phosphatase, CK, fibrinogen, GLU, and phosphorus (higher in young animals) and for globulin, Mg, and total protein (higher in adults). The RIs described here are invaluable for assessing and monitoring the health status of individual animals and herds. Our study highlights the importance of considering gender and age in the interpretation of biochemical parameters, providing critical insights for the conservation and management of this endangered breed.

## 1. Introduction

In Portugal, according to the *Official Catalog of Portuguese Autochthonous Breeds*, there are two autochthonous breeds of donkeys, Graciosa’s donkey and Miranda’s donkey, both with a very small number of pure specimens [1]. The population of Miranda’s donkeys consist of 761 animals, including 652 females and 109 males, distributed among 435 owners [2]. However, despite the increasing population trend, Miranda’s donkey is still considered an endangered breed [3].

Miranda’s donkey originated in the northern region of Portugal, specifically in Planalto Mirandês, where it adapted well due to its rusticity and resilience. It is characterized by having a dark brown coat with lighter gradations on the back; pronounced hirsutism with abundant hair on the back, face and extremities of the limbs; a height exceeding 130 cm; a voluminous head; short and thick neck; and large ears that are wide at the base [4].

For many years, the Miranda’s donkey breed was primarily employed for agricultural work, playing a vital role in the economic development of the region. However, with the onset of industrial mechanization, there has been a decline in the number of these animals [5,6]. To preserve this important Portuguese genetic and cultural heritage, several governmental and scientific incentives have been implemented. These include promoting the animals’ involvement in milk production, leisure activities, and asinotherapy [7,8,9].

Understanding the physiological and pathological conditions of Miranda’s donkey is essential to supporting the conservation strategies implemented for the breed. Several studies have been conducted on this breed, offering valuable insights to veterinarians across various domains, including physiology, reproductive pathologies, nutrition, parasitology, odontology, and hematology [10,11,12,13,14,15,16,17].

One crucial aspect of the gathered information involves establishing biochemical reference intervals, which is pivotal in characterizing the breed and assisting veterinarians in the interpretation of biochemical analysis results. This enables the assessment of the health status of individual animals or an entire herd, the determination of the severity and systemic effects of a disease, and the evaluation of treatment response [18,19,20]. A lack of well-defined biochemical reference ranges has been noted by researchers studying donkeys [21,22,23,24] and other autochthonous breeds [24,25]. To address the absence of specific information, veterinarians have traditionally relied on reference intervals (RI) from horses when interpreting results for donkeys due to their perceived similarities. However, this practice is strongly discouraged, as advancements in donkey medicine have demonstrated that these species exhibit several distinct characteristics [26]. Furthermore, various factors can exert an influence on serum biochemistry, including breed, gender, age, pregnancy, physical activity, season, and management practices [27,28,29,30].

Therefore, the aim of this study was to contribute to increasing knowledge on Miranda’s donkey physiology by establishing RIs for a set of important biochemical parameters and assessing the impact of gender and age on these measures.

## 2. Materials and Methods

### 2.1. Animals

This study was approved by the ORBEA (Ethics Committee for Animal Welfare) at Universidade Trás-os-Montes e Alto Douro (UTAD)-i467-e-CECAV-2022. All owners provided informed written consent for the use of their data and for the use of surplus blood samples after routine testing in further studies. In total, 75 clinically healthy Miranda’s donkeys were included in this study (33 males and 42 females, with an average age of 8.1 ± 5.7 years). All the animals were born and lived in the same geographical area (41°42′ N 06°48′ W, 652 m altitude) in the north of Portugal. The animals are registered with the Association for the Study and Protection of the Donkey Cattle (AEPGA—Associação para o Estudo e Proteção do Gado Asinino), and regularly monitored by the association’s veterinarians who assure their health status and oversee their nutritional and reproductive well-being. The veterinarians provide necessary care and treatments as needed, including regular prophylactic measures such as anthelminthic treatment [31].

The animals selected for the study followed specific inclusion criteria, which included only animals who were healthy, regularly dewormed, and had exhibited no signs of illness or health problems within the last 6 months. Owners or caretakers attested to the normal physical condition and regular activity of each animal. Pregnant and lactating mares were also included [18,25].

To study the effect of age on biochemical profiles, the donkeys were divided into two groups, young (1–3 years old) and adult (≥4 years old), following previous published studies [24,25]. Regarding the effect of gender, the donkeys were grouped into males and females.

### 2.2. Sample Collection and Biochemical Analyses

Sample collections were carried out at AEPGA in the morning period during the spring, as part of the annual prophylactic program tailored for the breed. All the animals were considered to be clinically healthy following anamnesis and physical examination by veterinarians, and were handled with care to minimize potential stress effects on the analyzed parameters.

Blood samples were obtained from non-fasted animals through puncture of the jugular vein with a 21-gauge needle (BD Vacutainer PrecisionGlide, Plymouth, UK) and added to 10 mL vacuum tubes with a pulverized coagulation activator (BD Vacutainer) for most of the biochemical parameters, and to a 4 mL tube containing citrate 3.2% (BD Vacutainer), respecting the 9:1 ratio of blood to citrate for fibrinogen analysis. Subsequently, the blood samples were subjected to routine centrifugation (800× *g* for 5 min) to obtain the serum and plasma. The serum and plasma were then transferred to identified aliquots and refrigerated for transport to the Veterinary Clinical Pathology laboratory at the Veterinary Hospital of University of Trás-os-Montes and Alto Douro (Vila Real) for analysis. According to laboratory standards, strongly hemolyzed or lipemic samples were rejected.

The following biochemical parameters were analyzed: Alkaline phosphatase (ALP), albumin (ALB), alanine aminotransferase (ALT), aspartate aminotransferase (AST), total bilirubin (TB), calcium (Ca), cholesterol (CHOL), creatine phosphokinase (CK), chlorine (Cl), creatinine (CREA), gamma-glutamyl transferase (GGT), globulin (GLO), glucose (GLU), potassium (K), phosphorus (P), magnesium (Mg), sodium (Na), total protein (TP), triglycerides (TRIG), and urea. These analyses were conducted using standard methods with an automated biochemistry analyzer (Respons^®^ 920, Diagnostic Systems GmbH, Holzheim, Germany) and reagent kits, following the manufacturer’s calibration and control standards. Fibrinogen (Fb) levels were assessed using semi-automatic equipment (Start^®^ 4-Diagnostica-Stago, Asnières sur Seine, France), also following the manufacturer’s calibration and control standards.

### 2.3. Statistical Analyses

For descriptive statistics, the reference interval (RI) and confidence interval (CI) were determined using a freely distributed application, Reference Values Advisor 2.1 [32]. This software employs the Anderson–Darling test to assess data normality and identifies outliers through the Tukey and Dixon–Reed methods. Depending on the distribution type, a parametric or robust method, with or without Box–Cox transformation, is applied. Subsequently, the RIs and 90% confidence intervals (CIs) are calculated for the lower and upper limits. The determination follows the guidelines of the ASVCP [33].

The analyses conducted to investigate potential significant differences regarding the impact of gender and age groups on biochemical parameters utilized SPSS software version 29. The Shapiro–Wilk and Kolmogorov–Smirnov tests were employed to assess sample normality, followed by subjecting the samples to an analysis of variance (ANOVA or Kruskal) [34]. Differences were considered significant when *p* < 0.05.

Animals that presented discrepant values, considered to be possible outliers, were excluded from the statistical analysis.

## 3. Results

### 3.1. Descriptive Analysis and Reference Intervals for the Total Number of Animals

Seventy-five healthy Miranda’s donkeys were included in this study, comprising 42 females with a mean age of 8 ± 6.09 years (ranging from 1 to 25 years) and 33 males with a mean age of 8.23 ± 5.14 years (ranging from 1 to 18 years). A total of twenty-one biochemistry parameters were measured, and all data are presented as the mean, median, standard deviation (SD), minimum and maximum values (Min–Max), RI, 90% confidence interval (CI) for the lower limit, and 90% CI for the upper limit. The RIs for all animals are described in Table 1.

### 3.2. Reference Intervals for Females and Males and Gender Effect

The RIs for the 42 females and the 33 males are presented in Table 2 and Table 3, respectively. It was possible to evaluate the full set of biochemistry parameters studied (*n* = 21) in both groups. For all parameters, RIs with respective lower and upper reference limits were obtained and described. The values of P, TP, ALB, AST, ALP, Ca, CHOL, CREA, Mg, GLOB, and UREA followed a normal distribution. The values of GLU, GGT, ALT, TB, CK, TRIG, Na, K, Fb, and Cl did not follow the normal distribution model.

Regarding the impact of gender on the analyzed parameters, significant differences were observed between groups (females versus males). Specifically, the levels of CHOL, Cl, CREA, GLU, Na, and TRIG were higher in females. Conversely, the levels of AST, CK, GGT, K, Mg, and UREA were higher in males (Table 4).

### 3.3. Reference Intervals for Young and Adult Animals and Age Effect

We present the results for the biochemistry parameters across different age groups, specifically 20 young animals and 55 adult animals, in Table 5 and Table 6, respectively.

We successfully evaluated the complete set of biochemistry parameters (*n* = 21) for both groups. Reference intervals (RIs) with lower and upper reference limits were established for all parameters. However, for triglycerides (TRIG) in the young group, the statistical software could not compute the lower and upper reference limits.

The analysis showed that the values of P, TP, ALB, AST, ALP, Ca, CREA, UREA, CHOL, Mg, and GLOB were normally distributed. In contrast, the values of Fb, GLU, GGT, ALT, TB, CK, TRIG, Na, K, and Cl did not adhere to the normal distribution model.

Significant age-related differences were observed between the groups (young versus adult animals). The levels of ALP, CK, Fb, GLU, and P were higher in young animals, while the levels of GLOB, Mg, and TP were elevated in adult animals (Table 7).

## 4. Discussion

Autochthonous breeds often carry unique genetic traits and adaptations that have evolved over generations in response to specific local conditions. Preserving these breeds contributes to the overall genetic diversity within livestock populations [6]. The literature describes reference intervals for several biochemical parameters in donkeys of different breeds, such as Catalonian [35], Ragusana [18], Âne Normand, and Âne du Contentin [24], Pêga [27], Amiata [36], Balkan [20], Kyrgyz [37], and Martina Franca [25]; however, there is no information concerning the Miranda’s Donkey breed. The present study investigates, for the first time, the RIs of a large set of biochemical parameters in healthy Miranda’s donkeys.

Comparing our results with those of other studies is difficult because, as original data from other breeds are not available, it is impossible to test the statistical differences between populations. Therefore, we restricted ourselves to an approximate interpretation of the comparisons, which should be approached with caution. Other factors can also make comparisons between studies difficult, such as differences in the methodologies and equipment used in different laboratories [18,38]. Furthermore, factors such as geographic conditions, seasons, climate, physiological conditions, diet, or even breed may contribute to differences in results [18,30].

In our study, we observed that several analytes were influenced by gender and age, emphasizing the importance of establishing appropriate reference ranges in these cases. No significant differences were found for AST in relation to the age of the animals, although values were higher in young animals. However, a significant difference was found between genders, with males presenting higher values. This finding is consistent with observations in Catalan donkeys [35] and Ethiopian donkeys [39]. The higher values in males can be explained by the dual origin of AST, which is produced in the liver but also in muscle cells. Generally, males possess higher muscle mass, contributing to AST serum levels. Elevated serum levels of this enzyme can result from hepatocyte damage. However, as AST is not liver-specific, it is crucial to interpret its findings in conjunction with other enzymes like CK and ALT [40].

Alkaline phosphatase and GGT constitute hepatobiliary enzymes and are the main indicators of cholestasis, along with hyperbilirubinemia. In addition to cases of cholestasis, increased GGT may be associated with the intake of medications, for example, barbiturates, and intense physical exercise [40,41]. The average GGT values described in our study were close to those found in a study conducted on the Âne Contentin and Âne Normand breeds in France [24]. No significant differences were found in relation to age; only gender showed variations, with males presenting higher values.

Alkaline Phosphatase is present in various tissues, including liver and bone. In the liver, ALP is located in the superficial bile canalicular membranes. Increased serum ALP concentrations are commonly associated with cholestasis. The assessment of ALP, along with GGT, is generally adequate to identify the origin and significance of the elevation [24].

In healthy animals, serum ALP originates mainly from the liver and bones, resulting in higher levels in growing animals and a subsequent decrease with advancing age. This age-related difference was observed in our study, revealing significant variation in ALP levels, with higher ALP values in young animals. Similar trends of higher ALP values in in young animals have been documented in other breeds, such as Balkan [20], Amiata [36] and Martina Franca [25].

Bilirubin in donkeys presents lower values compared to that in horses, which corroborates the importance of using specific values for this species [26]. Compared to other donkey breeds, our values of TB were similar to those described for Ragusana breeds [18]. No significant differences were found in TB values related to gender and age.

In our study, CK showed significant differences related to gender and age, with males and young animals showing higher values, which was also observed in the Ragusana breed [18]. CK is present in negligible amounts in many body tissues and in high concentrations in skeletal and cardiac muscle [41]. The higher CK levels in young animals and males can be attributed to their greater skeletal muscle mass compared to females, who generally tend to be of smaller size. Additionally, older animals, with less muscle mass, exhibited lower CK levels. CK was the parameter that showed the most variation between studies [20,21,42,43].

The CREA levels were found to be slightly higher in females, a finding that raises questions about the potential influence of including pregnant females in our study. The impact of pregnancy on CREA levels among different autochthones breeds appears to vary; for example, the Amiata breed showed no significant differences in CREA levels [44], while an increase in CREA levels during the second quarter of pregnancy was observed in the Martina Franca breed [45]. These conflicting results underscore the need for further research to elucidate the effects of pregnancy on CREA levels and to understand the biological mechanisms at play.

UREA levels were also influenced by gender, with males exhibiting higher values. The levels reported in our study align with those previously documented in donkeys in France [24], suggesting a consistent gender-related pattern in UREA levels across different populations.

A significant difference was found for P related to age, with a decreased concentration in adult animals. Similar results were also reported in Catalan [35] and Ragusana [18] donkeys. It is possible that this decrease occurs as a result of a decline in bone metabolism in adults [40].

For Mg, significant differences were found associated to gender and age, with values being slightly higher in males and adults. Similar results were described in males of the Pêga breed [27] and adults of the Martina Franca and Pêga breeds [25,27].

No significant differences were found for Ca, which corroborates what was described in the breeds Âne Cotentin and Âne Normand [24], Ragusana [18], and Pêga [27]. The values described were also very similar between breeds.

The K values described in Miranda’s donkey were similar to those described in several other breeds [18,25,27]. We did not find significant differences between age groups, which disagrees with some studies that have found differences between ages [18,25]. A significant difference was found between genders, with males showing slightly higher values. It is important to remember that K is an important intracellular ion, but its free concentration in plasma is low, which can make its measurement less accurate, contributing to the differences among studies [41].

Chlorine was influenced only by gender, with a higher value in females. Another study found an influence of age on the values of this electrolyte in donkeys of the Martina Franca breed [25].

The TP and ALB values for all animals were similar to the values found in donkeys in the United Kingdom [21], Italy [25,43], and France [24]. There was no significant difference in TP in relation to gender, but there was an influence of age, with an increase with age. This corroborates the findings of Pitel et al. [24], Trimboli et al. [25], and Girardi et al. [27].

Total proteins are important molecules involved in various organic functions. In serum, the main proteins found are ALB, GLOB, and fibrinogen. Changes in the amount of blood proteins may occur due to dehydration, inflammation, specific losses of one or more proteins, and changes in their synthesis and excretion [40]. Gender and age did not affect ALB values, but GLOB was influenced by age, with slightly higher values in adults, which was also described in the study in France [24] and can be justified by the levels of immunoglobulins developed during the lives of the animals. Fibrinogen, an important positive acute-phase protein, exhibits an elevated plasma concentration during inflammation. In horses and cattle, its concentration may rise even before an increase in white blood cell count, serving as an early indicator of inflammation [40]. In our study, a significant difference was found between age groups, with the youngest showing slightly higher values.

Significant differences in glucose (GLU) levels were observed, influenced by both gender and age, with females and young animals exhibiting higher GLU values. Notably, elevated GLU levels in young animals have also been documented in other autochthonous breeds, including the Ragusana [18], Martina Franca [25], and Andalusian [46]. Regarding gender, our findings diverge from those reported by Mendoza and colleagues [46], which could be attributed to the inclusion of pregnant females in our analysis. Supporting this, a study by Gloria et al. [45] indicates an increase in serum glucose levels in pregnant females, providing a plausible explanation for the heightened GLU levels observed in the female group in our study.

Regarding TRIG and CHOL, significant differences were found in relation to gender, with females exhibiting higher values. The establishment of normal triglyceride ranges for donkeys is of great importance in the diagnosis of hyperlipidemia and hypertriglyceridemia [47]. Donkeys are more susceptible to hyperlipidemia compared to horses and ponies and have high mortality rates [41]. Generally, TRIG levels increase during periods of fasting, but since our samples were collected from non-fasting animals, the levels should not have been interfered with.

A limitation of the present study is the irregular distribution of animals across the various groups under consideration (female, male, young, and adult). However, considering the endangered status of the breed in question, we believe the sample size is sufficiently representative. Despite this, it is clear that further research is required. Future studies should aim to incorporate a larger and more balanced number of animals across the different demographic groups to enhance the robustness and generalizability of the findings.

## 5. Conclusions

Miranda’s donkey is a Portuguese breed that, despite an increase in its population in recent years, still remains in danger, requiring actions for its preservation. Our study established RIs for biochemistry in a healthy population of Miranda’s donkeys. Several parameters were influenced by gender and age, factors that veterinarians must consider when interpreting biochemistry results. Finally, we encourage further research to be conducted to acquire more information about the physiology of this breed.

## Figures and Tables

**Table 1 animals-14-00805-t001:** Biochemistry reference intervals for healthy Miranda’s donkeys (*n* = 75).

Analyte/Units	*n*	Mean ± SD	Median	Min–Max	RI	LRL 90% CI	URL 90% CI
ALB g/dL	75	3.0 ± 0.2	3.0	2.15–3.43	2.4–3.4	2.2–2.6	3.3–3.4
ALT U/L	72	4.8 ± 1.5	4.5	1.4–9.9	2.5–8.6	1.4–2.9	7.8–9.9
AST U/L	75	239.6 ± 47.0	234.7	150.8–367.3	152.7–364.9	150.8–178.5	315.4–367.3
ALP U/L	75	160.1 ± 54.9	151.5	62.1–398.4	77.8–293.1	62.1–98.1	243.3–398.4
Ca mg/dL	75	11.3 ± 0.7	11.3	9.03–13.07	9.4–12.8	9.0–10.2	12.3–13.1
CHOL mg/dL	75	75.2 ± 13.9	75.0	52.0–116.0	52.9–113.3	52.0–54.9	99.0–116.0
CK U/L	73	117.3 ± 34.7	110.6	55.3–201.9	57.5–197.6	55.3–75.8	182.7–201.9
Cl mmol/L	73	105.6 ± 2.7	105.4	96.0–112.8	99.4–111.6	96.0–101.5	109.7–112.8
CREA mg/dL	75	1.2 ± 0.2	1.2	0.7–1.82	0.7–1.7	0.7–0.8	1.6–1.8
Fb	71	2.1 ± 0.5	2.1	0.96–3.74	1.0–3.60	1.0–1.4	3.1–3.7
GGT U/L	74	24.9 ± 8.9	22.9	10.7–57.8	11.1–50.8	10.7–14.2	40.9–57.8
GLO g/dL	75	3.9 ± 0.5	3.9	2.53–4.83	3.0–4.8	2.5–3.2	4.7–4.8
GLU mg/dL	75	81.7 ± 15.6	78.9	55.5–130.1	58.2–118.5	55.5–61.3	111.3–130.1
K mmol/L	75	3.8 ± 0.5	3.9	2.13–4.82	2.4–4.8	2.1–2.9	4.5–4.8
Mg mg/dL	75	2.2 ± 0.3	2.3	1.62–2.96	1.6–2.9	1.6–1.9	2.7–3.0
Na mmol/L	73	135.1 ± 2.7	135.4	125.3–140.5	126.6–140.0	125.3–130.3	138.7–140.5
P mg/dL	75	3.5 ± 0.8	3.4	1.91–5.37	1.9–5.2	1.9–2.3	4.8–5.4
TB mg/dL	75	0.1 ± 0.0	0.1	0.05–0.17	0.1–0.2	0.1–0.1	0.2–0.2
TP g/dL	75	6.9 ± 0.5	6.9	4.92–7.87	5.3–7.8	4.9–6.1	7.6–7.9
TRIG mg/dL	73	115.9 ± 39.2	114.0	14–205	44.6–193.1	14.0–61.3	177.8–205.0
UREA mg/dL	75	27.7 ± 10.0	26.5	11.6–58.1	13.0–54.0	11.6–13.6	44.3–58.1

ALB: albumin; ALT: alanine aminotransferase; AST: aspartate aminotransferase; ALP: alkaline phosphatase; Ca: calcium; CHOL: cholesterol; CK: creatine phosphokinase; Cl: chlorine; CREA: creatinine; Fb: fibrinogen; GGT: gamma-glutamyl transferase; GLO: globulin; GLU: glucose; K: potassium; Mg: magnesium; Na: sodium; P: phosphorus; TB: total bilirubin; TP: total protein; TRIG: triglycerides; SD: standard deviation; Min: minimum; Max: maximum; RI: reference intervals; LRL: lower reference limit; URL: upper reference limit; CI: confidence interval.

**Table 2 animals-14-00805-t002:** Biochemistry reference intervals for healthy females of Miranda’s donkey (*n* = 42).

Analyte/Units	*n*	Mean ± SD	Median	Min–Max	RI	LRL 90% CI	URL 90% CI
ALB g/dL	41	3.0 ± 0.2	3.0	2.5–3.43	2.5–3.4	2.5–2.8	3.3–3.4
ALT U/L	40	4.4 ± 1.2	4.5	1.4–8.0	1.4–8.0	1.4–2.7	5.8–8.0
AST U/L	42	228.6 ± 43.1	227.4	150.8–340.6	151.0–337.4	150.8–173.7	293.9–340.6
ALP U/L	42	166.3 ± 59.9	159.8	79.5–398.4	81.0–389.6	79.5–103.8	247.8–398.4
Ca mg/dL	42	11.4 ± 0.7	11.4	9.03–12.75	9.1–12.7	9.0–10.5	12.3–12.8
CHOL mg/dL	42	78.3 ± 14.4	76.0	52.0–116.0	52.2–115.8	52.0–58.4	102.6–116.0
CK U/L	41	108.2 ± 31.9	102.2	55.3–193.5	55.4–192.6	55.3–72.2	171.3–193.5
Cl mmol/L	42	106.7 ± 2.4	106.5	101.6–112.8	101.7–112.7	101.6–103.4	110.5–112.8
CREA mg/dL	42	1.3 ± 0.2	1.3	0.77–1.82	0.8–1.8	0.8–0.9	1.6–1.8
Fb	41	2.1 ± 0.5	2.1	0.96–3.74	1.0–3.7	1.0–1.5	2.9–3.7
GGT U/L	41	22.9 ± 8.0	22.2	10.7–47.1	10.7–46.8	10.7–13.3	39.5–47.1
GLO g/dL	42	3.9 ± 0.4	3.9	3.22–4.6	3.2–4.6	3.2–3.2	4.4–4.6
GLU mg/dL	42	88.4 ± 16.2	83.9	55.5–130.1	56.4–129.1	55.5–68.5	112.4–130.1
K mmol/L	41	3.8 ± 0.4	3.8	2.55–4.53	2.6–4.5	2.6–3.2	4.3–4.5
Mg mg/dL	42	2.2 ± 0.3	2.2	1.62–2.89	1.6–2.9	1.6–1.8	2.5–2.9
Na mmol/L	42	135.8 ± 2.2	135.7	126.8–139.6	127.3–139.6	126.8–133.8	138.6–139.6
P mg/dL	42	3.5 ± 0.7	3.5	1.91–4.83	1.9–4.8	1.9–2.0	4.7–4.8
TB mg/dL	42	0.1 ± 0.0	0.1	0.05–0.17	0.1–0.2	0.1–0.1	0.1–0.2
TP g/dL	42	6.9 ± 0.4	7.0	5.37–7.62	5.4–7.6	5.4–6.4	7.4–7.6
TRIG mg/dL	42	138.5 ± 61.4	128.0	14–356	17.6–353.9	14–65.3	190.9–356.0
UREA mg/dL	42	24.2 ± 9.6	23.4	11.6–58.1	11.7–57.8	11.6–13.8	38.0–58.1

ALB: albumin; ALT: alanine aminotransferase; AST: aspartate aminotransferase; ALP: alkaline phosphatase; Ca: calcium; CHOL: cholesterol; CK: creatine phosphokinase; Cl: chlorine; CREA: creatinine; Fb: fibrinogen; GGT: gamma-glutamyl transferase; GLO: globulin; GLU: glucose; K: potassium; Mg: magnesium; Na: sodium; P: phosphorus; TB: total bilirubin; TP: total protein; TRIG: triglycerides; SD: standard deviation; Min: minimum; Max: maximum; RI: reference intervals; LRL: lower reference limit; URL: upper reference limit; CI: confidence interval.

**Table 3 animals-14-00805-t003:** Biochemistry reference intervals for healthy males of Miranda’s donkey (*n* = 33).

Analyte/Units	*n*	Mean ± SD	Median	Min–Max	RI	LRL 90% CI	URL 90% CI
ALB g/dL	33	3.0 ± 0.2	3.0	2.39–3.39	2.4–3.4	2.1–2.6	3.3–3.5
ALT U/L	32	5.2 ± 1.7	4.9	2.7–9.9	2.6–9.4	2.3–3.1	8.0–11.0
AST U/L	33	253.7 ± 48.7	246.7	181.5–367.3	175.8–376.6	165.9–189.3	333.2–415.7
ALP U/L	33	152.3 ± 47.5	140.9	62.1–265.1	70.3–269.6	56.2–85.1	231.6–301.8
Ca mg/dL	33	11.2 ± 0.8	11.1	9.48–13.07	9.7–12.8	9.4–10.0	12.4–13.3
CHOL mg/dL	33	71.2 ± 12.3	71.0	53.0–97.0	49.4–99.9	45.9–53.5	92.3–108.3
CK U/L	32	128.9 ± 35.2	117.0	74.6–201.9	71.2–218.1	65.4–79.7	186.2–246.6
Cl mmol/L	31	104.1 ± 2.5	104.4	96.0–108.5	99.2–109.6	97.7–101.1	107.8–111.3
CREA mg/dL	33	1.2 ± 0.2	1.1	0.7–1.65	0.7–1.7	0.5–0.8	1.6–1.8
Fb	32	2.4 ± 0.8	2.3	0.97–4.8	1.1–4.7	1.0–1.3	3.7–5.8
GGT U/L	33	27.4 ± 9.6	25.3	12.0–57.8	14.5–55.5	12.8–16.5	42.6–73.3
GLO g/dL	33	3.8 ± 0.5	3.7	2.6–4.9	2.5–2.9	2.5–2.9	4.6–5.3
GLU mg/dL	32	72.4 ± 8.6	69.9	58.5–88.2	56.3–92.8	54.1–59.3	86.3–98.0
K mmol/L	33	4.0 ± 0.5	4.0	2.47–4.82	2.7–4.8	2.1–3.1	4.7–5.0
Mg mg/dL	33	2.3 ± 0.3	2.4	1.62–2.96	1.7–2.9	1.6–1.9	2.7–3.0
Na mmol/L	31	134.2 ± 3.2	134.4	125.3–140.5	126.9–140.2	124.5–129.3	138.6–141.6
P mg/dL	33	3.5 ± 0.9	3.2	2.21–5.37	1.5–5.2	1.1–1.9	4.5–5.7
TB mg/dL	33	0.1 ± 0.0	0.1	0.07–0.15	0.1–0.2	0.1–0.1	0.1–0.2
TP g/dL	33	6.8 ± 0.6	6.8	4.92–7.87	5.4–7.9	4.9–5.8	7.6–8.2
TRIG mg/dL	33	100.9 ± 30.5	98.0	50.0–165.0	39.3–165.8	26.3–52.7	148.5–182.6
UREA mg/dL	33	32.3 ± 8.6	33.4	13.5–47.1	11.6–48.1	5.2–18.1	44.9–50.5

ALB: albumin; ALT: alanine aminotransferase; AST: aspartate aminotransferase; ALP: alkaline phosphatase; Ca: calcium; CHOL: cholesterol; CK: creatine phosphokinase; Cl: chlorine; CREA: creatinine; Fb: fibrinogen; GGT: gamma-glutamyl transferase; GLO: globulin; GLU: glucose; K: potassium; Mg: magnesium; Na: sodium; P: phosphorus; TB: total bilirubin; TP: total protein; TRIG: triglycerides; SD: standard deviation; Min: minimum; Max: maximum; RI: reference intervals; LRL: lower reference limit; URL: upper reference limit; CI: confidence interval.

**Table 4 animals-14-00805-t004:** Effect of gender on biochemistry parameters in Miranda’s donkey.

Effect of Gender			
Analyte/Units	Females (*n* = 42)	Males (*n* = 33)	
	* Mean ± SD	* Mean ± SD	*p*-value
AST U/L	228.6 ± 43.1	253.7 ± 48.7	0.021
CHOL mg/dL	78.3 ± 14.4	71.2 ± 12.3	0.026
CREA mg/dL	1.3 ± 0.2	1.2 ± 0.2	0.036
Mg mg/dL	2.2 ± 0.3	2.3 ± 0.3	0.026
UREA mg/dL	24.2 ± 9.6	32.3 ± 8.6	<0.001
	** Median ± SD	** Median ± SD	*p*-value
GLU mg/dL	83.9 ± 16.2	69.9 ± 8.6	<0.001
GGT U/L	22.2 ± 8.0	25.3 ± 9.8	0.022
CK U/L	102.2 ± 31.9	117.0 ± 35.2	0.008
TRIG mg/dL	128.0 ± 61.4	98.0 ± 30.5	0.002
Na mmol/L	135.7± 2.2	134.4 ± 3.2	0.017
K mmol/L	3.8 ± 0.4	4.0 ± 0.4	0.013
Cl mmol/L	106.5 ± 2.4	104.1 ± 2.5	<0.001

AST: aspartate aminotransferase; CHOL: cholesterol; CK: creatine phosphokinase; Cl: chlorine; CREA: creatinine; GLU: glucose; GGT: gamma-glutamyl transferase; K: potassium; Mg: magnesium; Na: sodium; TRIG: triglycerides; SD: standard deviation. * Mean values were used for parameters with normal distribution; ** Median values were used for parameters with non-parametric distribution.

**Table 5 animals-14-00805-t005:** Biochemistry reference intervals for young healthy Miranda’s donkey (*n* = 20).

Analyte/Units	*n*	Mean ± SD	Median	Min–Max	RI	LRL 90% CI	URL 90% CI
ALB g/dL	20	3.0 ± 0.3	3.0	2.5–3.43	2.4–3.5	2.3–2.6	3.4–3.7
ALT U/L	20	5.5 ± 1.8	4.9	2.7–9.9	2.4–10.3	2.0–3.0	8.1–12.2
AST U/L	20	257.3 ± 47.8	262.7	172.8–344.5	154.3–361.6	126.6–191.2	329.6–392.9
ALP U/L	20	199.0 ± 44.0	198.1	109.8–281.4	106.8–296.4	75.8–136.0	261.4–324.9
Ca mg/dL	20	11.4 ± 0.6	11.4	10.47–12.56	10.2–12.7	10.0–10.6	12.3–13.1
CHOL mg/dL	20	80.1 ± 13.7	78.0	53.0–103.0	50.7–109.5	41.8–59.7	99.8–118.2
CK U/L	20	139.0 ± 35.7	139.9	87.4–201.9	62.4–215.6	38.3–89.6	192.3–241.3
Cl mmol/L	19	105.4 ± 1.7	104.9	102.8–108.9	101.6–109.2	100.6–102.8	107.9–110.3
CREA mg/dL	20	1.2 ± 0.3	1.2	0.7–1.82	0.6–1.9	0.4–0.7	1.7–2.2
Fb	19	2.3 ± 0.4	2.3	1.7–3.06	1.5–3.2	1.2–1.7	2.9–3.4
GGT U/L	20	26.0 ± 9.4	24.5	11.1–47.1	10.2–50.0	8.0–13.8	40.2–59.7
GLO g/dL	20	3.7 ± 0.4	3.7	3.07–4.3	2.9–4.5	2.6–3.1	4.3–4.7
GLU mg/dL	20	89.9 ± 15.6	86.1	67.3–117.2	55.2–124.7	48.6–65.2	112.7–132.8
K mmol/L	20	3.9 ± 0.2	3.9	3.62–4.38	3.6–4.5	3.6–3.7	4.3–4.8
Mg mg/dL	20	2.1 ± 0.2	2.2	1.62–2.51	1.6–2.6	1.3–1.8	2.5–2.7
Na mmol/L	20	134.9 ± 2.0	134.7	130.3–139.1	130.6–138.9	129.2–132.2	137.3–140.3
P mg/dL	20	4.3 ± 0.6	4.4	2.97–5.37	2.8–5.6	2.4–3.3	5.2–6.0
TB mg/dL	20	0.1 ± 0.0	0.1	0.07–0.14	0.0–0.1	0.0–0.1	0.1–0.2
TP g/dL	20	6.7 ± 0.3	6.6	6.12–7.47	6.0–7.5	5.9–6.2	7.2–7.9
TRIG mg/dL	20	112.7 ± 45.4	119.0	50.0–191.0	13.9–212.8	*	*
UREA mg/dL	20	27.9 ± 11.5	24.6	15.2–58.1	14.4–66.2	13.0–16.4	41.6–102.5

ALB: albumin; ALT: alanine aminotransferase; AST: aspartate aminotransferase; ALP: alkaline phosphatase; Ca: calcium; CHOL: cholesterol; CK: creatine phosphokinase; Cl: chlorine; CREA: creatinine; Fb: fibrinogen; GGT: gamma-glutamyl transferase; GLO: globulin; GLU: glucose; K: potassium; Mg: magnesium; Na: sodium; P: phosphorus; TB: total bilirubin; TP: total protein; TRIG: triglycerides; SD: standard deviation; Min: minimum; Max: maximum; RI: reference intervals; LRL: lower reference limit; URL: upper reference limit; CI: confidence interval; * non-computable by the system.

**Table 6 animals-14-00805-t006:** Biochemistry reference intervals for adult healthy Miranda’s donkey (*n* = 55).

Analyte/Units	*n*	Mean ± SD	Median	Min–Max	RI	LRL 90% CI	URL 90% CI
ALB g/dL	54	3.0 ± 0.2	3.0	2.39–3.39	2.5–3.4	2.4–2.7	3.3–3.4
ALT U/L	52	4.5 ± 1.3	4.5	1.4–8.3	1.8–8.2	1.4–2.8	6.4–8.3
AST U/L	55	233 ± 45.5	227.5	150.8–367.3	151.6–366.2	150.8–178.3	305.5–367.3
ALP U/L	54	141.3 ± 38.8	134.1	62.1–242.1	68.6–232.5	62.1–98.1	208.5–242.7
Ca mg/dL	55	11.2 ± 0.8	11.2	9.03–13.07	9.2–12.9	9.0–10.1	12.3–13.1
CHOL mg/dL	55	73.4 ± 13.6	73.0	52.0–116.0	52.4–114.8	52.0–55.0	97.1–116.0
CK U/L	53	109 ± 30.9	104.2	55.3–193.5	56.2–189.9	55.3–73.5	173.4–193.5
Cl mmol/L	53	105.8 ± 3.0	105.7	96.0–112.8	97.7–112.3	96.0–101.7	110.4–112.8
CREA mg/dL	55	1.2 ± 0.2	1.2	0.77–1.67	0.8–1.7	0.8–0.9	1.6–1.7
Fb	52	2.1 ± 0.6	2.0	0.96–3.74	1.0–3.7	1.0–1.3	3.0–3.7
GGT U/L	53	26.9 ± 22.5	22.7	10.7–178.2	11.2–133.3	10.7–14.6	40.8–178.2
GLO g/dL	55	3.9 ± 0.5	3.9	2.53–4.83	2.8–4.8	2.5–3.2	4.7–4.8
GLU mg/dL	55	78.7 ± 14.6	77.0	55.5–130.1	56.7–123.1	55.5–60.7	102.0–130.1
K mmol/L	55	3.8 ± 0.6	3.9	2.13–4.82	2.3–4.8	2.1–2.8	4.5–4.8
Mg mg/dL	55	2.3 ± 0.3	2.3	1.62–2.96	1.7–2.9	1.6–1.9	2.8–3.0
Na mmol/L	53	135.2 ± 3.0	135.7	125.3–140.5	125.8–140.3	125.3–130.5	139.2–140.5
P mg/dL	55	3.2 ± 0.6	3.2	1.91–4.72	1.9–4.6	1.9–2.1	4.0–4.7
TB mg/dL	55	0.1 ± 0.0	0.1	0.05–0.17	0.1–0.2	0.1–0.1	0.2–0.2
TP g/dL	54	7.0 ± 0.5	7.0	5.37–7.87	5.6–7.8	5.4–6.3	7.6–7.9
TRIG mg/dL	53	117.2 ± 37.0	114.0	14.0–205.0	29.1–199.8	14.0–66.6	181.8–205.0
UREA mg/dL	55	27.7 ± 9.5	28.3	11.6–47.1	12.2–46.2	11.6–13.6	43.4–47.1

ALB: albumin; ALT: alanine aminotransferase; AST: aspartate aminotransferase; ALP: alkaline phosphatase; Ca: calcium; CHOL: cholesterol; CK: creatine phosphokinase; Cl: chlorine; CREA: creatinine; Fb: fibrinogen; GGT: gamma-glutamyl transferase; GLO: globulin; GLU: glucose; K: potassium; Mg: magnesium; Na: sodium; P: phosphorus; TB: total bilirubin; TP: total protein; TRIG: triglycerides; SD: standard deviation; Min: minimum; Max: maximum; RI: reference intervals; LRL: lower reference limit; URL: upper reference limit; CI: confidence interval.

**Table 7 animals-14-00805-t007:** Effect of age on biochemistry parameters in Miranda’s donkey.

Effect of Age			
Analyte/Units	Young (*n* = 20)	Adult (*n* = 55)	
	* Mean ± SD	* Mean ± SD	*p*-value
ALP U/L	199.0 ± 44.0	141.3 ± 38.8	<0.001
GLO g/dL	3.7 ± 0.4	3.9 ± 0.5	0.019
Mg mg/dL	2.1 ± 0.2	2.3 ± 0.3	0.041
P mg/dL	4.3 ± 0.6	3.2 ± 0.6	<0.001
TP g/dL	6.7 ± 0.3	7.0 ± 0.5	0.025
	** Median ± SD	** Median ± SD	*p*-value
Fb	2.3 ± 0.4	2.0 ± 0.6	0.026
GLU	86.1 ± 15.6	77.0 ± 14.6	0.005
CK	139.9 ± 35.7	104.2 ± 30.9	0.005

ALP: alkaline phosphatase; CK: creatine phosphokinase; Fb: fibrinogen; GLO: globulin; GLU: glucose; Mg: magnesium; P: phosphorus; TP: total protein; SD: standard deviation. * Mean values were used for parameters with normal distribution; ** Median values were used for parameters with non-parametric distribution.

## Data Availability

All the data supporting the results are included in the manuscript. The dataset is available from the corresponding author on reasonable request.

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
