# Peer review of "Serum Biochemistry Parameters of the Endangered Miranda’s Donkey Breed: Reference Intervals and the Influence of Gender and Age"

_animals, 2024, doi:10.3390/ani14050805_

Round 1

Reviewer 1 Report

Comments and Suggestions for Authors

lines 72-74: Donkeys present relevant anatomical and physiological differences within different breeds. This field has been investigated deeply in many different countries, consider adding the following references supporting your research and providing more background info: Radiological anatomy of the donkey's foot: Objective characterisation of the normal and laminitic donkey foot (Collins et al., 2011); Radiographic parameters of the digit in a cohort population of Amiata donkeys (Nocera, I., et al., 2020); Evaluation of different methods to estimate the transfer of immunity in donkey foals fed with colostrum of good IgG quality: A preliminary study (Turini, L, 2021); Blood analysis in newborn donkeys: Hematology, biochemistry, and blood gases analysis (Veronesi et al., 2014)

lines 105-107: the study population was grouped according to age, could you please add references to support your choice, since equids 1-3 y.o. might be considered adults from a physiological point of view?

Please consider adding "limits" paragraph for your research paper.

Comments on the Quality of English Language

Author Response

Comments and Suggestions for Authors

Referee comment: lines 72-74 Donkeys present relevant anatomical and physiological differences within different breeds. This field has been investigated deeply in many different countries, consider adding the following references supporting your research and providing more background info: Radiological anatomy of the donkey's foot: Objective characterisation of the normal and laminitic donkey foot (Collins et al., 2011); Radiographic parameters of the digit in a cohort population of Amiata donkeys (Nocera, I., et al., 2020); Evaluation of different methods to estimate the transfer of immunity in donkey foals fed with colostrum of good IgG quality: A preliminary study (Turini, L, 2021); Blood analysis in newborn donkeys: Hematology, biochemistry, and blood gases analysis (Veronesi et al., 2014).

Authors answer: The authors thank for the suggestions. Due to the fact that we are considering only biochemical RI values, the work by Turini and colleagues was added to the reference list. The article by Veronesi et al. 2014 was already included in the previous version of our manuscript.

lines 105-107: The study population was grouped according to age, could you please add references to support your choice, since equids 1-3 y.o. might be considered adults from a physiological point of view?

Authors answer: Authors followed previous published works. The references were included in the material and methods section.

Referee comment: Please consider adding "limits" paragraph for your research paper.

Authors answer: Authors followed referee suggestion and included a paragraph about study limitations at the end of discussions section.

Reviewer 2 Report

Comments and Suggestions for Authors

Dear authors,

The article is interesting and provides valuable data about biochemistry parameters of the endangered Miranda’s donkey breed. I have some minor considerations (see the comments below).

Results

I suggest that you move the column with RI in Tables 1,2,3,5 and 6 to the end of the table considering that RI is the final goal of this study.

In Table 4 - check the significance for chlorides, if the count is missing or less than 0.001.

Did all the variables in Table 4 and 7 have a normal distribution given that you reported all the values as Mean ± SD.

Line 168: "UREIA" replaced with "UREA"

Discussion

Lines 273-234: The authors reported that the creatinine level was higher in females than in males. This is an interesting finding, considering that CREA was found to be higher in males [22 and 27], with the explanation that this differentially stems from greater muscle mass in males. Given that you included pregnant mares in the study, could that be the explanation for your CREA results (see ref. 24). The same goes for glucose.

Lines 288-290: I cannot agree with this statement for Mg considering that there is a study [27], that showed a similar result as yours. Please correct it.

Lines 323-324: I did not find that the mentioned authors [18, 24, 43] compared glucose between the sexes and reported increased values in females, except for [40], who did not find a difference. Please correct it.

Author Response

Referee 2:

Comments and Suggestions for Authors

Referee comment: The article is interesting and provides valuable data about biochemistry parameters of the endangered Miranda’s donkey breed. I have some minor considerations (see the comments below).

Authors answer: The authors thank for the comments.

Results

Referee comment:

I suggest that you move the column with RI in Tables 1,2,3,5 and 6 to the end of the table considering that RI is the final goal of this study.

Authors answer:

The referee is right, the RI is the final goal of our study. However, the columns (LRL 90% CI) and (URL 90% CI), represent respectively the Lower and the Upper reference limits for the RI, therefore, in the authors point of view, should be mentioned after the RI values.

Referee comment: In Table 4 - check the significance for chlorides, if the count is missing or less than 0.001.

Authors answer: The referee is right. It was a typing error corrected in this new version (p <0.001).

Referee comment: Did all the variables in Table 4 and 7 have a normal distribution given that you reported all the values as Mean ± SD.

Authors answer: The authors followed the referee's suggestion and created a variation of previous table 4 and 7, to include the requested information.

Referee comment: Line 168: "UREIA" replaced with "UREA".

Authors answer: The referee is right. It was a typing error corrected in this new version

 Discussion

Referee comment: Lines 273-234: The authors reported that the creatinine level was higher in females than in males. This is an interesting finding, considering that CREA was found to be higher in males [22 and 27], with the explanation that this differentially stems from greater muscle mass in males. Given that you included pregnant mares in the study, could that be the explanation for your CREA results (see ref. 24). The same goes for glucose.

Authors answer: Authors thank the referee's suggestion. The text was changed (lines 292-303)and additional discussion was also added for glucose (lines 340-348).

Referee comment: Lines 288-290: I cannot agree with this statement for Mg considering that there is a study [27], that showed a similar result as yours. Please correct it.

Authors answer: Authors thanks the referee suggestion. The text was changed.

Referee comment: Lines 323-324: I did not find that the mentioned authors [18, 24, 43] compared glucose between the sexes and reported increased values in females, except for [40], who did not find a difference. Please correct it.

Authors answer: Authors thanks the referee's suggestion. The text was changed and references were updated (lines 340-348).

Reviewer 3 Report

Comments and Suggestions for Authors

Dear Authors,

Thanks for the work performed in your manuscript entitled "Serum biochemistry parameters of the endangered Miranda's donkey breed: Reference intervals and the influence of gender and age"

Please find below my suggestions to improve your manuscript:

Please verify throughout the document : for, from, I think there are some English editing that could be improve in this preposition.

L22: add clinical before the word results

L26: remove or justify " due to the small number of specimens". Readers know what an endanger specie is, there is not need to explain it or please write down if you know how much percentage of the breed has been reduced.

Abstract: Please verify English language in the abstract. 

L29: remove For this, before "blood samples from....

L31: RI was already explained, no need to write down the complete word

L32: The version of the SPSS software in the abstract seems unnecessary

Material and methods

Animals: L88 could you please add the range of average of the individuals used in this study? for example, X number of donkeys ranging X age

L105: could you please explain the use of mares in this study? what was the purpose? to compare against jenny's data/results, I suppose?

Results

L156: please verify time tense in the "The RI for all animals are described" instead of were I think

Influence of gender 

L165: I think you could expand more in the explanation of the results part. For example, you could explain if females selected in this study were from the 1st or 2nd parturition or at which state of the reproduction period were they approximately and why were they selected at this particular state (with a reference)

L190 Influence of age

Could you please expand and explain the criteria selection on the animal's age...why did you selected this age, probably to reduce pathologies related to the young age of the donkeys....to help them improve and understand their immunity system evolution?

And in the adults Table 6, please describe when a donkey is consider an adult, expand...not as an introduction or a big paragraph but to expand and explain your criteria selection.

L234 please remove " and we will discuss some of them".

L245 which medication increases GGT?

L275 and 278 please rephrase to clarify this paragraph could be improved

Could you please just add a short paragraph about how all of this parameters could influence on the animal's health and drug effect if there is something available in donkeys/horses or other equid specie.

Thanks in advance,

With kind regards,

Reviewer

Author Response

Referee 3:

Referee comment: Thanks for the work performed in your manuscript entitled "Serum biochemistry parameters of the endangered Miranda's donkey breed: Reference intervals and the influence of gender and age".

Please find below my suggestions to improve your manuscript:

Please verify throughout the document: for, from, I think there are some English editing that could be improve in this preposition.

Referee comment: L22: add clinical before the word results.

Authors answer: Done.

Referee comment: L26: remove or justify " due to the small number of specimens". Readers know what an endanger specie is, there is not need to explain it or please write down if you know how much percentage of the breed has been reduced.

Authors answer: Done

Abstract

Referee comment: Please verify English language in the abstract.

Authors answer: The abstract was revised.

Referee comment: L29: remove For this, before "blood samples from....

Authors answer: Done.

Referee comment: L31: RI was already explained, no need to write down the complete word.

Authors answer: Done.

Referee comment: L32: The version of the SPSS software in the abstract seems unnecessary.

Authors answer: Done, information removed.

Material and methods

Referee comment: Animals: L88 could you please add the range of average of the individuals used in this study? for example, X number of donkeys ranging X age

Authors answer: information was included (line 90)

Referee comment: L105: could you please explain the use of mares in this study? What was the purpose? to compare against jenny's data/results, I suppose?

Authors answer: The decision was made considering the methodology used in similar studies on other autochthonous breeds (the references were cited at the end of the sentence). It was not the objective of this study to investigate the differences in biochemical values between pregnant and non-pregnant females. However, we agree that such a study will be very useful and should be done in a next future.

Results

Referee comment: L156: please verify time tense in the "The RI for all animals are described" instead of were I think

Authors answer: Done.

Influence of gender

Referee comment: L165: I think you could expand more in the explanation of the results part. For example, you could explain if females selected in this study were from the 1st or 2nd parturition or at which state of there production period were they approximately and why were they selected at this particular state (with a reference).

Authors answer: We expanded the information in the results. We don’t have in our files the requested specific information related to the 1st of 2nd parturition of females or at which stage of their production period just because this was not one aim of our study. We hope the referee understands. In the Material and Methods section a reference is already allocated to justify the followed methodology.

Influence of age

Referee comment: L190: Could you please expand and explain the criteria selection on the animal's age...why did you selected this age, probably to reduce pathologies related to the young age of the donkeys....to help them improve and understand their immunity system evolution?

And in the adults Table 6, please describe when a donkey is consider an adult, expand...not as an introduction or a big paragraph but to expand and explain your criteria selection.

Authors answer: The choice of age range was based on other work carried out with donkeys (references 24 and 25). The information requested is already included in the material and methods section. Although we understand the referee's point of view, it seems to us unnecessary to duplicate the information about methodology in the results section. The text was slightly expanded.

Referee comment: L234 please remove " and we will discuss some of them".

Authors answer: Done.

Referee comment: L245 which medication increases GGT?

Authors answer: The drug is barbiturates. Information was included in the text.

Referee comment: L275 and 278 please rephrase to clarify this paragraph could be improved.

Authors answer: The authors changed this sentence.

Could you please just add a short paragraph about how all of these parameters could influence on the animal's health and drug effect if there is something available in donkeys/horses or other equid specie.

Authors: We did not understand this last referee request. Our apologies for that.

Round 2

Reviewer 1 Report

Comments and Suggestions for Authors

All concerns was addressed, thus the paper might be accepted in present form